# Immunomodulatory Precision: A Narrative Review Exploring the Critical Role of Immune Checkpoint Inhibitors in Cancer Treatment

**DOI:** 10.3390/ijms25105490

**Published:** 2024-05-17

**Authors:** Junyu Qiu, Zilin Cheng, Zheng Jiang, Luhan Gan, Zixuan Zhang, Zhenzhen Xie

**Affiliations:** 1College of Basic Medical, Nanchang University, Nanchang 330006, China; alvinqjy@163.com (J.Q.); czl1303561190@163.com (Z.C.); 13865755166@163.com (Z.J.); glh18779981646@163.com (L.G.); 15826774576@163.com (Z.Z.); 2Queen Mary School, Medical Department, Nanchang University, Nanchang 330031, China; 3Huan Kui School, Medical Department, Nanchang University, Nanchang 330031, China

**Keywords:** immune checkpoint inhibitors (ICIs), immune response, combination immunotherapies, cancer treatment, anti-cancer drugs

## Abstract

An immune checkpoint is a signaling pathway that regulates the recognition of antigens by T-cell receptors (TCRs) during an immune response. These checkpoints play a pivotal role in suppressing excessive immune responses and maintaining immune homeostasis against viral or microbial infections. There are several FDA-approved immune checkpoint inhibitors (ICIs), including ipilimumab, pembrolizumab, and avelumab. These ICIs target cytotoxic T-lymphocyte-associated protein 4 (CTLA-4), programmed cell death protein 1 (PD-1), and programmed death ligand 1 (PD-L1). Furthermore, ongoing efforts are focused on developing new ICIs with emerging potential. In comparison to conventional treatments, ICIs offer the advantages of reduced side effects and durable responses. There is growing interest in the potential of combining different ICIs with chemotherapy, radiation therapy, or targeted therapies. This article comprehensively reviews the classification, mechanism of action, application, and combination strategies of ICIs in various cancers and discusses their current limitations. Our objective is to contribute to the future development of more effective anticancer drugs targeting immune checkpoints.

## 1. Introduction

Substantial progress has been made in the field of cancer immunotherapy due to the development of immune checkpoint inhibitors (ICIs), which have demonstrated high efficacy. ICIs are prominent tumor therapies due to their broad biological activity in various histological tumor types, resulting in long-lasting antitumor effects [1,2]. Most current ICIs target the molecules CTLA-4, PD-1, and PD-L1. Ipilimumab, a CTLA-4 blocker, is the first FDA-approved drug targeting an immune checkpoint to treat melanoma [3]. PD-1 and PD-L1 can be considered a pair of factors; PD-L1 is overexpressed on the tumor cell surface and interacts with PD-1, leading to T-cell apoptosis and immune escape in patients with cancer [4]. PD-1/PD-L1 inhibitors prevent this binding and restore T-cell function [5]. Currently, ICIs are being evaluated in clinical trials for various types of tumors, including melanoma [6,7], non-small cell lung cancer (NSCLC) [8,9], renal cell carcinoma (RCC) [10], uroepithelial cancer (UC) [11], and head and neck squamous cell carcinoma (HNSCC) [12] (Table 1). The Nobel Prize in Physiology or Medicine was awarded to James P. Allison and Tasuku Honjo in 2018 for their work on ICIs and the significant role of these drugs in tumor suppression.

ICIs have a wider range of applications than traditional treatments and provide longer-lasting therapeutic responses. Current research on tumor immunotherapy emphasizes the need to investigate synergistic interactions among multiple ICIs. ICIs are well suited for combating drug resistance and enhancing the ability of immune cells to identify and eliminate tumor cells through combination therapies that target multiple pathways. For example, combining CTLA-4 inhibitors with PD-1 inhibitors may improve the prognosis of lung cancer patients and prolong their survival. Similarly, combining ICIs with various cancer therapies is considered a promising strategy. For instance, the combination of anti-VEGF antibodies with PD-1 antibodies has been confirmed to be highly effective in treating advanced NSCLC and urothelial cancers [13].

This review focuses on the specific molecular mechanisms of three immune checkpoint molecules and their various applications, including monotherapy or combination strategies, to explore the potential of ICIs in future cancer treatment. Moreover, we selected five cancers, namely, four solid tumors and lymphomas, all of which have shown benign responses to ICI strategies, and discussed the specific ICI applications and underlying problems for these tumors in detail. Comparing the advantages of ICIs with those of traditional tumor treatments and combining these treatments where appropriate is a worthwhile consideration. In addition, this review considers the limitations of ICIs and presents new insights into their development prospects.

## 2. Immunoregulation and Mechanisms of Action of Immune Checkpoint Inhibitors

### 2.1. Immune Checkpoints and Their Role in Immunoregulation

Immune checkpoint molecules on the surface of immune cells, particularly T cells, play a pivotal role in regulating the immune system [14]. These molecules downregulate immune responses and prevent abnormal immune events and autoimmunity by establishing “tolerance” [15]. These molecules can be stimulatory, such as CD27, CD40, and CD137, or inhibitory, such as CTLA-4, PD-1, KIR, IDO, LAG3, and NOX2 [14,16]. This review focuses on inhibitory molecules. Naïve T cells possess a specific T-cell receptor (TCR), which is a crucial marker for T cells. The TCR targets epitopes presented by antigen-presenting cells (APCs) through major histocompatibility complex (MHC) molecules. Dendritic cells (DCs) are typical APCs in diverse peripheral tissues that phagocytize exogenous cells. Subsequently, APCs present antigens to T cells for antigen recognition via MHC molecules. MHC I and II molecules are the two main classes involved in immune recognition. Antigens on MHC-I interact with cytotoxic T cells, while those on MHC-II interact with helper T cells, both of which activate naïve T cells. However, activating T cells alone is insufficient for their complete function. Costimulatory molecules are required to induce an effective immunogenic response. These molecules can generate stimulatory or inhibitory signals, and the final effects depend on the stronger signaling component. T cells can only be fully activated through two-signal communication involving TCR–MHC and costimulatory molecule interactions. This process is commonly referred to as the “two-signal model of activation” [17,18]. Without these costimulatory signals, T cells might become anergic to stimulation or even undergo apoptosis [19,20].

The immune system plays a crucial role in various diseases, particularly in tumor surveillance and clearance. Although, in most cases, the immune system recognizes cancer cells as “enemies”, tumor cells may undergo specific changes in their characteristics to evade immune defense. Tumor cells express immune checkpoint molecules to downregulate immune responses and establish “tolerance”, inhibiting T cells’ normal activation and maturation [2,21]. PD-1/L-1, CTLA-4, and CD80/CD86 interactions are two crucial immune checkpoint pathways in the tumor microenvironment (TME). In recent years, there has been a growing emphasis on investigating the inhibition of these interactions. Various monoclonal antibodies (mAbs) have been developed as “powerful weapons” for blocking this process; ICIs, for example, have shown durable and enhanced therapeutic effects in various clinical conditions and experiments [22]. At present, several ICIs have been approved by the FDA for the clinical treatment of various cancers, including nivolumab and pembrolizumab for blocking PD-1, atezolizumab and avelumab for blocking PD-L1, and ipilimumab and tremelimumab for blocking CTLA-4 on the cell surface [2]. Although clinical trials have revealed some negative conditions, such as low overall response rates, severe adverse reactions, and tumor regression, ICIs are still being thoroughly investigated and explored for their potential to improve therapeutic effects [23,24].

### 2.2. Classification of ICIs

ICIs play a crucial role in the process of T cell-tumor interactions by blocking the connection between immune checkpoint receptors and their corresponding ligands. Research in this field primarily focuses on PD-1/PD-L1 inhibitors and CTLA-4 inhibitors due to their widespread use in clinical trials. Furthermore, this review considers cytokine-induced SH2 protein (CISH), a new type of adoptive immunotherapy (Table 2). The mechanisms of these immune checkpoint inhibitors (ICIs) are diverse but share commonalities, which will be demonstrated below (Figure 1).

#### 2.2.1. PD-1 and PD-L1 Inhibitors

Over the last decade, immunotherapies, especially PD-1/PD-L1 inhibitors, have achieved unprecedented success. PD-1 is mainly expressed on T cells but can also be expressed on several lymphocytes. PD-1’s ligands, PD-L1 or PD-L2, are typically expressed on the surface of APCs and regulate immune responses or on the surface of tumor cells to evade immune defense. In most cases, T cells are prevented from proliferating, activating, or secreting cytokines after interacting with their receptors and ligands [14,16]. However, on the surface of certain tumor cells with specific mutations, PD-1 ligands are overexpressed and bind to the corresponding T cells, resulting in T-cell exhaustion or inactivation. PD-L1 inhibitors have been developed to restore normal T-cell function and reinvigorate T cells in the tumor microenvironment. In recent years, this advanced immunotherapy has shown remarkable and promising results [34], with high survival rates and durable remission in patients with melanoma, NSCLC, and some mismatch repair-deficient tumors. The PD-L1-related miRNA profile has been shown to have the potential to predict the response of lung squamous cell carcinoma (LUSC) patients to PD-L1/PD-1 inhibitors, helping identify the optimal treatment cohort [35]. PD inhibition-based combination therapies are considerably more effective than PD-1 blockade monotherapy [26,36], indicating that combination treatment is promising.

During T-cell activation, PD-1 communicates with PD-L1 or PD-L2, causing a change in its conformation and the phosphorylation of its cytoplasmic tail by Src family kinases [37,38,39]. This binding affects two major signaling pathways—PTEN-PI3K-Akt and RAS-MEK-ERK [40,41]—through the recruitment of the phosphatase SHP-2. While PD-1 is often associated with T-cell exhaustion and tumor defense, it is not exclusive to exhaustion. In contrast, PD-1 is not present in naïve or resting T cells but is found in all T cells during activation, making it a biomarker of effector T cells [14,42,43]. Upon prompt removal of the activating antigen, PD-1 levels decrease in T cells that are responsive to the stimulus. However, in cases where the antigen persists, such as in cancer or chronic infections, PD-1 levels may remain elevated and sustained. Accumulating evidence has shown that high expression levels of PD-L1 in the TME are positively associated with poor prognosis and survival [44,45]. Additionally, tumor-associated macrophages (TAMs), as major components of the TME, significantly impact the therapeutic efficacy of PD-1/PD-L1 inhibitors [46].

As described above, blocking the PD-1/PD-L1 pathway has been extensively investigated in recent years since many tumor types express PD-L1 and manifest T-cell dysfunction [47]. To date, several drugs targeting the PD-1 pathway, including the mAbs nivolumab, pembrolizumab, atezolizumab, avelumab, and durvalumab, have been approved by the FDA to treat various cancers. Although different response rates have been recorded for diverse cancer types, determining the specific biological characteristics of the drug before its actual application to patients facilitates better outcomes. Additionally, there is a positive correlation between the expression level of PD-L1 and the possibility of a response to anti-PD therapy strategies in melanoma patients [48,49]. Notably, the expression of PDL1 in the TME cannot be considered the only biomarker for the selection or exclusion of patients for anti-PD therapies. Some patients who do not express PD-L1 still demonstrate objective responses to such therapies [14]. The effectiveness of anti-PD therapies in producing clinical responses is also impacted by the presence of intratumoral T-cell infiltration or Th1 cell gene expression [50,51]. However, while some tumors are heavily infiltrated by T cells, others are not, which may be linked to the migration of dendritic cells and subsequent activation of T cells [52]. Epigenetic silencing of the tumor Th1-type chemokines CXCL9 and CXCL10 also contributes to severe T-cell infiltration and greater tumor evasion [53]. Researchers have also noted that higher mutation burdens could predict better anti-PD therapy efficacy, as “nonself” immunogenic antigens from somatic mutations may induce host T-cell reactions [54]. PD-L1 expression and tumor mutational burden are common predictors of the response to anti-PD-1/PD-L1 therapy in lung cancer, but other factors, such as tumor-specific genes, dMMR/MSI, and the gut microbiome, also show promise as predictive biomarkers. Noninvasive peripheral blood biomarkers, including DNA-related biomarkers and hematological cell-related biomarkers, are being utilized to predict immunotherapy responses [55]. The gut microbiota plays a significant role in the efficacy of anti-PD-1/PD-L1 antibodies in patients with colorectal cancer, with evidence suggesting that altering the gut microbiota composition can enhance antibody effectiveness. Moreover, the strength of the bond between a chimeric antigen receptor (CAR) and its target antigen impacts how responsive CAR-T cells are to inhibition by PD-1/PD-L1. The PD-1/PD-L1 axis inhibits T cells in CAR-T-cell therapy for solid tumors, but disrupting this pathway is complex. One study revealed that the affinity between a CAR and its antigen is crucial for determining how susceptible T cells are to PD-1/PD-L1 inhibition [56], particularly in diseases such as lung cancer [57].

The success of combining biological research and clinical application is evident in new immunotherapies. PD inhibition-based combination therapies are considerably more effective than PD-1 blockade monotherapy [26,36], indicating that combination treatment is promising. Neoadjuvant PD-(L)1 inhibitors are safe and effective for treating muscle invasive bladder cancer. Combining these agents with other ICIs and chemotherapy can lead to higher response rates but more severe side effects [58]. ICI therapies demonstrate notable limitations, such as a lack of response in most patients and severe side effects and adverse reactions in several systems after therapy. Resistance to ICIs is primarily associated with dysregulation of antigen presentation and interferon-γ signaling pathways. This effect highlights the crucial role of the TME in the development of resistance, underscoring the importance of studying molecular mechanisms at the cellular level.

Adverse reactions are relatively common after PD-1/PD-L1 inhibitor therapy. Approximately 9.47% of patients treated with PD-1/PD-L1 inhibitors experienced varying degrees of diarrhea, which is the most common adverse reaction to PD-1/PD-L1 and is usually grade 1–2. Common endocrine system immune-related adverse reactions include hypothyroidism (6.07%; 95% CI, 5.35–6.85%) and increased AST (3.39%; 95% CI, 2.94–3.89%). Among others [59], hyperthyroidism (2.82%; 95% CI, 2.40–3.29%) was the other most prevalent endocrine system immune-related adverse reaction. Deaths in patients receiving anti-PD-1 or anti-PD-L1 antibodies were attributed primarily to pneumonia (115 of 333 (35%)), hepatitis (75 of 333 (22%)), and neurotoxic effects (50 of 333 (15%)).

#### 2.2.2. Cytotoxic T Lymphocyte-Associated Protein 4 (CTLA-4) Inhibitors

CTLA-4 is a coinhibitory molecule expressed on the surface of T cells and a member of the CD28 immunoglobulin subfamily. The main ligands of CTLA-4 are CD80 and CD86, which are expressed on both APCs and tumor cells. The binding of these ligands to CTLA-4 can activate or inhibit immune responses. The PI3K/Akt pathway shows an overlap in signaling points between CD28 and CTLA-4, indicating their crucial regulatory role. CTLA-4 maintains T-cell homeostasis and tolerance through intracellular and extracellular pathways. The inhibitory effect of CTLA-4 is primarily mediated through competitive interaction with CD28 [60]. Moreover, its intrinsic signaling pathway is thought to regulate CTLA-4 localization and T-cell motility, thus indirectly activating T cells [61,62]. CTLA-4 is one of the most recognized immune checkpoints and has been widely used in cancer immunotherapy.

The CTLA-4-encoding gene is located on chromosome 2q33, adjacent to the gene encoding the CD28 receptor [63]. Although both receptors bind to CD80 and CD86, they demonstrate opposite functions. CD28 bolsters T-cell activation, while CTLA-4 inhibits T-cell activation [30,31]. CD80 and CD86 have greater affinities for CTLA-4 than for CD28, resulting in the inhibition of T-cell activation by CD28 binding [32]. Traditional naïve CD4+ and CD8+ T cells express CTLA-4 on their surface only after activation. TCR-mediated T-cell stimulation occurs before these naïve T cells are induced, and these cells act as inhibitory molecules against CD28 [64]. The complete mechanisms underlying the inhibitory function of CTLA-4 remain unclear; however, several studies suggest that this molecule affects the downstream activity of CD28 and TCR [65]. When CTLA-4 competes with CD28 for binding to CD80 or CD86, a series of cell activities, such as T-cell proliferation and survival, are inhibited. However, unlike CD28, CTLA-4 does not persist on the cell surface; CD28 only moves to the surface of the intracellular cytoplasm upon T-cell activation. Additionally, clusters of CTLA-4 arise where CD28 is present; thus, CD28 can be excluded from the binding site [66,67]. The extracellular structures of CD28 and CD80/86 are quite similar. However, steric interference occurs in most membrane-proximal domains of the CD28 dimer, reducing the affinity between CD28 and CD80/86 versus CD28 and CTLA-4 [33]. In addition to competing with CD28, CTLA-4 exerts an inhibitory effect by mediating the transendocytosis of CD80/86, meaning that it can transport CD80/86 from APC surfaces for degradation in CTLA-4-expressing cells [68]. This type of inhibition in traditional T cells is not as effective as that in Treg cells, which constantly express CTLA-4 on their surface [69]. The activation of Treg cells via CTLA-4-independent immunosuppression can impair antitumor immunity mediated by CTLA-4 blockade. The depletion of Treg cells in the TME using anti-CTLA-4 mAbs with antibody-dependent cell-mediated cytotoxicity (ADCC) activity is a key mechanism for achieving tumor regression [70]. Significantly, in vitro research has shown that CTLA-4 is expressed not only on T cells but also on DCs, which affects many DC activities [71]. DCs can also express a soluble form of CTLA-4 that influences surrounding DCs’ functions, indicating a decrease in CD80/86 on APCs [72,73]. There is a notable increase in CTLA-4 expression within cancer cells, accompanied by elevated Treg expression in the adjacent TME, which helps these cells evade immune surveillance [74]. The FDA previously approved several ICIs targeting CTLA-4, including the mAb ipilimumab for melanoma worldwide and renal cell carcinoma in the US [75,76] and tremelimumab for diverse types of cancers [77]. Increasing evidence suggests that combining CTLA-4 inhibitor immunotherapy and traditional therapies may yield superior outcomes and reduce drug resistance reported for monotherapies [28,78]. In addition, immune-related adverse reactions, including colitis, hepatitis, dermatitis, thyroiditis, and hypophysitis, have been reported in over 60% of patients receiving CTLA-4-based immunotherapies [79]. Thus, some patients may benefit from personalized treatment strategies targeting CTLA-4. In clinical studies, CTLA4 deficiency rescued functional T cells in patients with leukemia who had failed previous CAR-T-cell therapy. Thus, selective CTLA4 deficiency may revitalize functionally impaired T cells in patients with chronic lymphocytic leukemia (CLL), providing a strategy to enhance patient responsiveness to CAR-T-cell therapy [80].

Immune-related adverse events (irAEs) are more frequent with anti-CTLA-4 antibodies (e.g., navulizumab or pabolizumab) than with anti-PD-1/PD-L1 antibodies and demonstrate a different spectrum of organ involvement. As many as 60% of patients treated with ipilimumab experience any-grade irAE, of which 10–30% are generally considered serious (defined as grade 3–4 according to the National Cancer Institute’s Common Terminology Criteria for Adverse Events (CTCAE)) [81]. In that analysis, colitis was the most common cause of death arising from an irAE in patients receiving anti-CTLA-4 antibodies (135 of 193 deaths (70%)), and the combination of anti-CTLA-4 and anti-PD-1 antibodies increased the incidence and severity of irAEs [82].

#### 2.2.3. Cytokine-Induced SH2 Protein

The CISH gene encodes the cytokine-induced SH2 protein, a member of the suppressor of cytokine signaling (SOCS) family. This protein plays a crucial role in activating T cells, NK cells, and cytokine-induced proliferation, making it a significant checkpoint in both innate and adaptive immune responses [83]. The activity of T cells is intricately regulated, which presents a balance between Teff and Treg. Impaired Treg promotes the activity of self-reactive effector T cells and leads to autoimmunity. In this circumstance, cytokine signaling (especially IL-2) is essential for T cell homeostasis, and here CISH acts as a feedback inhibitor [84]. Tumor cell growth can be accelerated by negative regulation, inhibitory ligands, and blunted T-cell signals within the TME [85]. To enhance antigen reactivity and avoid peripheral tolerance, many efforts have been made to increase TCR signaling strength and generate highly functional T cells. However, strategies to bypass tolerance and increase avidity showed time consumption and off-target toxicities [86,87]. CISH expression is also associated with a longer metastasis-free interval (MFI) in triple-negative breast cancer (TNBC) and can refine the prognostic value of PD-L1 expression. This finding may underscore the clinical relevance of combining CISH inhibition with anti-PD-1/PD-L1 [88]. Therefore, CISH knockout has emerged as a novel approach to improving functional avidity, and there is growing interest in its involvement in immune regulation [89,90].

Studies have shown that CISH plays a vital role in T-cell function in allergic, infectious diseases, and malignancies, as well as in the activation and differentiation of naïve T cells [91,92]. CISH was validated for its association with non-functional tumor-infiltrating lymphocytes (TIL). A recent clinical trial showed that CISH disruption via CRISPR/Cas9 technology could increase metastatic gastrointestinal tumor vulnerability to ICIs by enhancing TIL neoantigen recognition in tumor cells [89,90]. The experiment involved transferring tumor-specific T cells with disrupted CISH to tumor-bearing mice. The results showed an overall increase in T cell activity towards neoantigens and a positive response to PD-1 inhibitor treatment in vivo. Additionally, there was a negative correlation between high CISH levels and TIL activities in the patients. It is important to note the pronounced synergistic treatment effect from the combination of PD-1 inhibitors and CISH-knocked TILs. The findings demonstrate that the removal of internal checkpoints, combined with external checkpoint blocking, synergistically improves TIL sensitivity to tumor neoantigens, resulting in enhanced therapeutic efficacy [89,90,91,92]. This combination of ICIs and adoptive cellular transfer offers promising possibilities for tumor treatment.

Previous studies indicated that CISH is expressed after TCR stimulation and then binds to PLC-γ1, resulting in its degradation and regulation of early T-cell signaling [91]. T cells in the antigen-relevant TME may upregulate CISH and manifest worse CD8+ T-cell functions [91]. CISH gene deletion enhanced CD8+ T-cell expansion, functional avidity, and cytokine release. In addition, CISH-deficient CD8+ T cells adoptively metastasize into poorly immunogenic tumors with profound and long-lasting regression [91]. The adoptive transfer involved tumor-specific CD8+ T cells distinguished by both high functional avidity and deficient or knocked-down CISH. This strategic approach resulted in a substantial and enduring regression of poorly immunogenic, established tumors, and the observed outcomes hold promise for advancing immune checkpoint therapies. Additionally, researchers demonstrate a critical role for CIS in suppressing natural killer (NK) cell-mediated control of tumor initiation and metastasis. Mice lacking CISH are highly resistant to methylcholanthrene-induced sarcoma and prevent lung metastasis of B16F10 melanoma and RM-1 prostate cancer cells. Combining CISH knockout with targeted therapies such as BRAF and MEK inhibitors, immune checkpoint blockade antibodies, IL-2, and type I interferons reveals further control of metastasis. These data suggest that targeting CIS can enhance the anti-tumor function of NK cells, and CIS holds great promise as a new target for NK cell immunotherapy.

As an intrinsic checkpoint inhibitor of TCR signaling and tumor immunity [91], the role of CISH in human anticancer immunity has been inadequately documented. Future research is anticipated to provide more clues in this regard, potentially offering new avenues for ICIs and adoptive cell therapies.

## 3. Applications of ICI Therapy in Various Types of Cancer

In recent years, ICI has garnered escalating attention as a leading immunotherapy. As a result, a significant amount of research on it has been conducted in the field of cancer treatment. For the best efficacy in different types of cancer, ICIs are designed with specific biomarkers and their unique mechanisms and applied to respective tumors. Here we take melanoma, lung cancer (especially NSCLC), renal cell carcinoma (RCC), bladder cancer, and lymphoma as examples to illustrate the diverse clinical applications of ICIs and some challenges that appeared during treatment (Figure 2).

### 3.1. Melanoma

Melanoma is an aggressive and life-threatening skin cancer often caused by excessive exposure to UV light, which can lead to a high tumor mutation burden (TMB). Melanoma is highly immunogenic, which contributes to its malignancy. Given the evidence of T lymphocyte infiltration into the TME and the occurrence of vitiligo, melanoma elicits strong immune responses, indicating that autoimmune responses defend melanocytes [93,94]. However, melanoma is highly susceptible to metastasis, suggesting that immunosuppression or dysfunction may counteract its immunogenicity.

Despite the strong immunogenicity of melanoma, it is not completely cleared by immune surveillance. Melanoma-infiltrating T cells are enriched in tumor-specific antigens, meaning that they have fully matured. However, recent evidence suggests that these TILs may be ineffective in eradicating tumor cells. Nonetheless, TILs could still provide valuable information for the development of tumor vaccines or adoptive cell transfer (ACT) [95]. The elevation of Treg and PD-1/PD-L1 communication in the TME may strongly establish “tolerance” for melanoma, accompanied by dampened T-cell activation, elevated exhaustion, and inhibited TIL function [96,97,98]. Melanoma-infiltrating Treg cells are highly clonal and can recognize tumor cells via TCR and MHC-II communication, indicating that melanoma cells can activate and amplify Tregs to modulate the surrounding immunosuppressive environment [99].

The conditions above indicate that melanoma can activate unique programs to prevent immune clearance, offsetting active immune responses. This immune-evasive nature of tumor progression highlights the vulnerability of melanoma to immunotherapies, explaining the responsiveness of melanoma to ICIs [95]. Ipilimumab, a mAb functioning as a CTLA-4 inhibitor, was initially approved for treating metastatic melanoma in 2011 [75]. Subsequently, pembrolizumab and nivolumab were approved in 2014 and were shown to improve overall survival in patients with metastatic melanoma compared with that achieved with ipilimumab or chemotherapy [100,101,102,103,104,105]. Recently, LAG-3 has emerged as a promising ICI-targeting biomarker and is being evaluated in patients with melanoma. LAG-3 is another surface inhibitory receptor upregulated in antigen-stimulated T cells, such as Tregs [106]. Given the different and complementary effects of ICIs, a combination treatment strategy is preferred. An exemplary approach is the combination of CTLA-4 and PD-1, which has demonstrated prolonged efficacy in metastatic melanoma compared to that achieved with individual therapies. Approximately 50% of treated patients survive beyond 6.5 years [107]. Obtaining additional information about tumor genomes and relevant biomarkers may lead to the development of more effective therapies, especially for melanoma, which has a high frequency of somatic mutations, resulting in a notable abundance of neoantigens. The challenges mentioned above may soon be resolved with the widespread use of next-generation sequencing (NGS) [97]. Moreover, understanding the mechanisms of toxicity generation in ICI-related therapies remains crucial, benefiting the management of melanoma and other tumor types. Overall, the clinical success of ICIs in treating melanoma has confirmed the therapeutic impact of reinvigorating the immune system. However, half of the patients treated with ICIs or ICI combination therapies experience adverse reactions [107].

### 3.2. Lung Cancer (NSCLC)

NSCLC is a typical type of lung cancer characterized by high levels of somatic nonsynonymous mutations (also known as TMB), especially in metastatic lung cancers [108,109]. Mutations inevitably introduce neoantigens, initiating T-cell antitumor responses. However, immunosuppression or checkpoint inhibition in TILs may inhibit this defense mechanism. Significant progress in ICI development has instigated a paradigm shift in patients with lung cancer, especially those with locally advanced NSCLC without EGFR/ALK alterations, for whom ICIs have been approved as second-line therapy after numerous trials [9,110,111,112]. In comparison to traditional chemotherapy, ICI treatment is associated with long-term survival in a fraction of patients with advanced NSCLC. The use of classical ICIs, such as nivolumab and pembrolizumab, increases the overall survival rate of patients with previously treated advanced NSCLC [113,114]. However, this effect is observed in only a small fraction of patients with high PD-L1 expression [115]. Thus, ICI combination strategies are preferred due to their unique and complementary mechanisms for more durable and profitable effects. The combination of atezolizumab (a PD-L1 inhibitor) with chemotherapy for the treatment of extensive-stage small cell lung cancer (ES-SCLC) has been found to increase overall survival rates compared with those observed with chemotherapy alone [116]. Moreover, the simultaneous inhibition of CTLA-4 and PD-L1, as exemplified by the combination of nivolumab and ipilimumab, has been investigated as a first-line therapy for patients with advanced NSCLC lacking EGFR/ALK alterations. This approach significantly increased overall survival rates compared with those observed with chemotherapy, irrespective of PD-L1 expression levels [117]. SCLC, accounting for approximately 15% of lung cancers, is noted for its high lethality and poor prognosis. Treatment strategies for SCLC have significantly lagged those for NSCLC. Despite trials incorporating a combination of ICIs such as ipilimumab and bevacizumab with traditional platinum-based regimens, no evident clinical benefit was observed in most [118,119]. Encouragingly, the success of the IMpower133 study represents a promising breakthrough in the treatment of SCLC. This study evaluated the first-line combination of atezolizumab and etoposide and demonstrated a significant improvement in overall survival rates despite more irAEs [116]. Given the findings of the IMpower133 study, the FDA has approved first-line atezolizumab combined with chemotherapy for patients with SCLC.

Notably, individuals with EGFR/ALK alterations have demonstrated minimal therapeutic benefit with ICI monotherapy. This observation may be attributable to the limited immunogenicity associated with such alterations [120,121]. In addition, it seems that ICI-based therapies had worse effects in advanced NSCLC patients with brain or liver metastasis [122,123]. To date, PD-L1 and TMB are two well-accepted biomarkers. Also, many others are pointed out as biomarkers for predicting the effects of such immunotherapies, while there is still no appropriate biomarker for ICI applications. A recent study highlighted a significant association between PD-L1 expression and biopsy site [124]. Additionally, another study reported inconsistencies in the calculation method used to evaluate TMB, casting doubt on its potential as a biomarker [125]. Hence, a comprehensive understanding of biomarkers holds paramount significance in the selection of ICIs. Acquired resistance is common in ICI-related therapies due to complex cancer-immune interactions within the tumor, TME, and host immunity. A deeper understanding of these biological mechanisms is crucial to overcome challenges and improve long-term survival for lung cancer patients.

### 3.3. Renal Cell Carcinoma (RCC)

The emergence of ICIs has revolutionized the treatment of advanced RCC. Historically, there has been a perceived absence of efficacious treatment strategies beyond surgical resection for RCC. However, unlike many other immune-responsive tumors, RCC exhibits a relatively high TMB [126], making it difficult to predict RCC patients’ clinical responses to ICIs. Clear-cell renal cell carcinoma (ccRCC) represents approximately 70% of all RCC cases and is characterized by higher immune infiltration than that observed in non-ccRCC subtypes [127]. ccRCC induces potent immunosuppression primarily by accumulating immune-inhibitory cells within the TME, such as Tregs and myeloid-derived suppressor cells. This accumulation inhibits the function of effector T cells and APCs by upregulating checkpoint molecule expression. Studies have indicated that the proliferative activity of CD8+ TILs is a more accurate predictor of prolonged patient survival than the mere abundance of infiltrating cells [128]. It has long been noted that increased infiltration of RCC is associated with poor prognosis, a pattern that contrasts with the prognosis trends observed in nearly all other solid tumors [129]. The underlying causes may include the misrecognition of tumor antigens, the high heterogeneity of infiltrating T cells, and metabolic alterations within these T cells in the RCC TME [130]. VEGFs can promote immunosuppression by enhancing the accumulation of Tregs in the TME and retarding DC maturation, in addition to accelerating angiogenesis [131]. ccRCC is identified as a highly angiogenic tumor characterized by alterations in the VHL gene and increased expression of VEGF-A compared to other tumor types [132]. However, the presence of increased TILs in RCC contrasts with preclinical evidence suggesting that elevated VEGF-A expression is typically associated with reduced immune infiltration [133,134]. The distinct TME generated by RCC could partially explain this phenomenon, and further exploration of additional factors is warranted [130]. Given the nature of RCC and VEGF, diverse immunotherapies have been developed and applied as primary therapeutic strategies.

The use of immunotherapy for RCC has changed dramatically over the last decade. Cytokine-based, high-dose interleukin-2 (IL-2) was initially used in some patients and achieved a durable complete response (CR) [135]. Inhibition of the VEGF pathway with a tyrosine kinase inhibitor (VEGF-TKI) or the anti-VEGF antibody bevacizumab subsequently emerged, demonstrating well-documented effects on T-cell infiltration into tumors and modulation of TIL functions in the TME [136,137]. Recently, ICI-based combinations (nivolumab plus ipilimumab) and ICIs combined with VEGF TKIs have shown remarkable efficacy in patients with metastatic RCC, and they have become the standard-of-care first-line therapies for patients with this disease. Notably, most patients develop primary or acquired resistance after the initial response [75,138]. Related adverse reactions are inevitably reported in patients who receive ICIs, possibly due to the nonspecific targeting of reinvigorated immune cells to normal tissues. New biomarkers, including TIM-3, a member of the immunoglobulin superfamily expressed on B cells and DCs, are continually being explored. In RCC, the expression of TIM-3 on PD-1-positive CD8+ T cells has been associated with a poor prognosis. This finding suggests a novel target for ICCs, although its expression may not be as ubiquitous as that of PD-1 on T cells [29,139]. Adoptive T-cell therapy, including the infusion of TILs expanded in vitro, has demonstrated success in treating melanoma but has yielded disappointing results in RCC [140,141]. The reasons for this discrepancy remain unclear and may be related to compromised T-cell functions. A 2018 study revealed that the immune response of TILs from patients with RCC was weaker than that of TILs from patients with melanoma [142].

In conclusion, more basal mechanism research and trials for novel combination therapy strategies are expected. The rapid development of immunogenomic tools, such as mass spectrometry and single-cell sequencing technologies, will now enable more systematic discovery of RCC tumor antigens. With consideration of the established immune responsiveness of RCC, it is promising to develop more satisfactory strategies to benefit RCC patients shortly.

### 3.4. Bladder Cancer

Bladder cancer is one of the most aggressive neoplasms worldwide; most patients (approximately 70%) present with less aggressive non-muscle-invasive bladder cancer (NMIBC), and the rest with muscle-invasive types (MIBC) that often progress to metastatic malignancy [143]. Additionally, bladder cancer has the highest mutation burden, followed by lung and skin cancer [144]. The TME consists of malignant stromal cells and TILs, in which cancer-associated fibroblasts (CAFs) are vital stromal cells and are strongly associated with the pathogenesis of many malignancies [145]. The ability of CAFs to immunomodulate and crosstalk with tumor cells has attracted attention; immunohistochemical (IHC) analyses of primary tumors from patients with bladder cancer revealed increased CAFs compared with that in normal bladder tissue [146]. The expression of the CAF markers CD90 and FAP is positively correlated with bladder cancer aggressiveness [147]. Additionally, the stromal expression of CAF markers strongly correlates with epithelial–mesenchymal transition (EMT) in cancer cells [148]. Furthermore, CAFs are associated with TILs, as noted in a previous study in which bladder cancer-infiltrated T cells accumulated in the stroma without interacting with tumor cells. This lack of interaction hinders effective immune clearance [149]. These findings indicate the unique role of the TME in bladder cancer invasion and evasion and the special role of stromal cells in this disease.

The rapid advancement of NGS has facilitated the identification of specific genetic variants and biomarkers, leading to the establishment of a molecular taxonomy of bladder cancer [150]. This progress has promoted novel therapeutic strategies for bladder cancer in recent years. Traditionally, cisplatin-based regimens have been employed as first-line therapies for metastatic urothelial carcinoma (mUC). However, the outcomes are unsatisfactory, as nearly all patients ultimately experience disease progression and succumb, even initial responders to treatment [151,152]. Intravesical Bacillus–Calmette–Guerin (BCG) was the first approved immunotherapy for bladder cancer and is still commonly applied. BCG works by inducing robust innate immune responses and long-lasting adaptive immunity, with approximately 40% of patients eventually relapsing despite an initial response [153,154]. The detection of high levels of immune checkpoint expression in bladder cancer patients is known to be associated with aggressive tumors, providing novel insights for immunotherapy development. The approval of several ICIs by the FDA, such as atezolizumab and nivolumab, for the treatment of MIBC patients represents a vital shift in the treatment of bladder cancer [155]. After that, atezolizumab and pembrolizumab were approved for first-line treatment of advanced cisplatin-ineligible bladder cancer patients with high PD-L1 expression [156]. However, only a small subset of patients can receive durable benefits from ICI monotherapy. Therefore, the exploration of combining ICIs with chemotherapy or other ICIs has been a persistent area of investigation. For advanced bladder cancer, the combinations of anti-PD-1 and anti-CTLA-4 seem to be particularly important and are under testing [157]. Treatment of NMIBC involves transurethral resection of the bladder tumor (TURBT), followed by intravesical chemotherapy or immunotherapy [158]. ICIs in treating BCG-refractory NMIBC are a hot topic of several ongoing trials [159], and combination strategies like anti-PD-1 along with TKI dasatinib were also identified as eligible for bladder cancer [160].

The role of immunotherapy has been strongly demonstrated at almost every stage of the disease, but it is explicit that chemotherapy remains the most favorable first-line treatment option. Research on the optimal first-line metastatic therapy is still evolving, as the balance between the benefits of immunotherapies and immune-related adverse reactions should be focused on. Standardized and reproducible biomarkers are in urgent need for clinical applications of novel ICIs, and the combination of ICIs and other compounds like FGFR inhibitors and anti-angiogenic drugs heralds promising therapeutic choices [161]. Biomarkers with predictive roles are expected for best patient selection, for the benefit of this widely distributed but intractable cancer.

### 3.5. Lymphoma

In addition to these solid tumors, ICIs have shown great promise in the treatment of lymphomas, a type of blood cancer that includes non-Hodgkin’s lymphoma (NHL) and Hodgkin’s lymphoma (HL) [162]. Lymphomas are unique in that the tumor cells originate from immune cells, making ICIs particularly effective due to their dual action on both the immune environment and the tumor cells themselves. Traditional treatments like chemotherapy and radiotherapy often lead to severe side effects and disease recurrence in lymphoma patients [163,164], underscoring the urgency of investigating ICIs in lymphoma.

Classical Hodgkin lymphoma (cHL), which constitutes over 95% of HL cases, features a small number of tumor cells derived from B-cells amidst a majority of non-malignant reactive cells [162]. These tumor cells, which typically lose their B-cell receptor, are susceptible to apoptosis and rely heavily on external signals for survival. This vulnerability is further exacerbated by mutations in the PD-L1 gene and alterations in chromosome 9p24.1, leading to increased PD-L1 and PD-1 in the tumor microenvironment (TME) via JAK-STAT signaling [165]. The amounts and distributions of PD-L1 expressed on tumor cells depend on lymphoma subtypes, underscoring that molecular assessment may help diagnose and predict therapeutic responses. In addition, human leukocyte antigen (HLA) is vital for tumor neoantigen presenting to induce antitumor responses. However, lymphomas can prevent these responses by expressing immune checkpoints [166,167]. A team described that aberrant membrane invariant chain peptide (CLIP) expression in HLA class II cell surface positive lymphoma cells, especially in cHL, can prevent the presentation of antigenic peptides to induce immune evasion, and an HLA expression pattern incompatible with normal antigen presentation is common in cHL, diffuse large B-cell lymphoma (DLBCL), primary central nervous system Hodgkin lymphoma (PCNSL), as well as testicular lymphoma [168]. Together, aberrant HLA expression should be focused on and considered when evaluating the effectiveness of ICIs in B-cell lymphomas.

In terms of therapeutic outcomes, ICIs targeting PD-1, PD-L1, and CTLA-4 have demonstrated efficacy in achieving prolonged survival in patients with aggressive lymphomas, even post-bone marrow transplant [169]. Among the various tumors expressing PD-L1, lymphoma is the most responsive to anti-PD therapies. Patients with refractory HL treated with nivolumab against PD-1 showed an overall response rate of more than 80% and also showed encouraging results in relapsed/refractory DLBCL with an overall response rate (ORR) of 36% [170,171]. Ipilimumab against CTLA-4 has been tested with rituximab in 33 patients with CD20-positive refractory B cell lymphoma by Tuscano et al., and about half of the patients had a response, particularly in follicular lymphoma patients [172]. So, further evaluation of diverse kinds of ICIs alone or in combination with other agents in B-cell lymphoma patients is promising [173]. Despite these encouraging results, complete remissions are rare, and related adverse reactions cannot be avoided [174]. It also remains to be established which patients might benefit most from checkpoint inhibition.

The effectiveness of ICIs in lymphomas presented the necessity of continuing research into these therapies, especially in less studied areas like NHL. The current understanding is significantly influenced by the interaction between lymphomas and their TME, and there is a pressing need for studies that can more closely mimic real-body situations due to challenges in sustaining primary tissue-derived Reed-Sternberg (RS) cells in culture [166]. These remind us of the necessity to explore ICI and related combination strategies in lymphomas.

## 4. Distinctions between Traditional Interventions and ICI Treatment (Table 3)

### 4.1. Surgery and ICI

There is a growing trend toward comparing operative techniques with ICIs in cancer management. Surgery has traditionally been the primary treatment option for early stage tumors [175]. However, the emergence of immunotherapy has broadened the range of options available to patients for controlling and managing their disease.

Surgical intervention involves removing malignant tumors or tissue. Although surgery can effectively remove all of the visible signs of cancer, it may also effectively eliminate any remaining cancer cells that have spread to other parts of the body [176]. Immune checkpoint inhibitors prevent cancer cells from sending ‘off’ signals by blocking checkpoint proteins from binding to their chaperones, allowing T cells to kill cancer cells. Clinical trials have shown that ICIs effectively treat various cancer types, and these drugs are generally well tolerated by patients despite some side effects [177].

Comparative analyses have revealed the efficacy of surgical procedures and ICIs for specific cancer varieties. For instance, one study focusing on patients with advanced melanoma revealed that the overall survival rate was more favorable among individuals treated with ICIs than among those who underwent surgery alone. Moreover, immunotherapy based on ICIs has revolutionized the management of metastatic and adjuvant melanoma, significantly extending disease-free survival (DFS) and progression-free survival (PFS) times in treated individuals [178].

Whether to apply surgical intervention or ICIs as efficacious therapeutic options for malignant tumors depends on numerous considerations, including the tumor type, cancer stage, and the patient’s general health condition. ICIs are potentially promising approaches for treating cancer. Further scientific inquiries are expected to provide deeper insights into the best methods for cancer management and treatment.

### 4.2. Chemotherapy and ICI

Chemotherapy and ICIs are two distinct modalities in the fight against cancer. Chemotherapy has long been a standard treatment option. ICIs are a more recent form of immunotherapy and have shown promising clinical outcomes.

Chemotherapy drugs are usually administered intravenously and can cause side effects such as hair loss, nausea, and fatigue [179]. Chemotherapy effectively reduces tumor size and destroys cancer cell structure. However, a major problem in cancer treatment is targeting chemotherapy drugs at malignant cells without affecting benign cells [180]. The effectiveness of certain chemotherapeutics, such as antimetabolites [181] and topoisomerase inhibitors [182], depends on the extent to which the drugs disrupt the rapid proliferation of cancer cells during the mitotic phase of their cell cycle [180]. Various complex factors can interfere with this process.

ICIs block the signals that cancer cells use to evade the immune system, allowing the body’s natural defense mechanisms to identify and eliminate cancer cells more efficiently [183]. In contrast to chemotherapy, ICIs selectively target specific immune response pathways involved in cancer cell elimination, mitigating numerous adverse effects.

Regarding efficacy, chemotherapy can induce rapid and significant tumor shrinkage. However, this effect may be temporary, as cancer cells can develop sustained resistance to chemotherapy drugs over time. This resistance can be attributed to the immunological phenomenon in which the immune system retains a memory of cancer cells, reducing ongoing protection against malignancy [184].

### 4.3. Targeted Therapy and ICI

Targeted therapy is a precision medicine approach that uses pharmaceuticals or therapeutic modalities to specifically target molecules or pathways responsible for cellular growth, proliferation, and dissemination. In the field of cancer treatment, this method aims to interfere with cancer cells and specific biological pathways involved in tumor expansion and progression. This is accomplished by identifying molecular targets, such as molecules associated with apoptosis and angiogenesis, that are observed to be either overexpressed or mutated in cancerous tissues. As a result, the intervention leads to the suppression of malignant cells [185].

When comparing targeted therapy with ICIs, several factors come into play. Targeted therapies are tailored to the number of specific genetic alterations found in a patient’s cancer cells. This personalized approach directly targets specific changes that drive cancer growth, helping to improve the effectiveness of treatment [186]. In contrast, ICIs are more effective in activating the immune system to target cancer cells, irrespective of their genetic mutations.

Another crucial factor to consider is the potential adverse effects associated with these interventions. While targeted therapy has the potential to cause side effects such as skin rash, inflammation of the nail folds, or reactions in the digestive tract [187], ICIs may elicit immune-related side effects, including inflammation in various organs. Moreover, the nature and severity of the side effects may also vary among individuals. Furthermore, the response rates to these treatments may display variability. Targeted therapies typically elicit a prompt response, with tumors beginning to shrink within a few weeks of initiating treatment. Conversely, ICIs may require a longer duration for response assessment as they work by stimulating the immune system to identify and attack cancer cells [188].

Both therapies have their advantages and limitations. When deciding which treatment is best for an individual’s cancer diagnosis, the potential side effects and response rates should be considered.

### 4.4. Radiation Therapy and ICI

Radiation therapy (RT) is a cancer treatment method that uses high-energy radiation, such as X-rays and gamma rays, to target and kill cancer cells and shrink tumors. RT damages cancer cell DNA, preventing cell division and growth [189]. RT can be used alone or in combination with other therapies.

A significant difference between RT and ICIs is their respective mechanisms of action. RT acts as a promoter that induces immunity in situ by killing tumor cells and triggering a systemic immune response [190]. ICIs are mAbs that target inhibitory checkpoint molecules expressed by APCs and CD4+ T-cell membranes, thereby modulating the immune system [183]. This distinction has implications for determining which treatment is most effective depending on the type of cancer. RT is particularly effective in controlling local tumors, making it suitable for solid tumors. In contrast, ICIs show great promise in treating metastatic cancer, where the tumor has spread to other parts of the body.

Furthermore, the side effects of these treatments differ. RT can cause temporary side effects, such as fatigue, skin changes, and hair loss, depending on the treated area [191]. In contrast, ICIs can cause autoimmune and inflammatory reactions, where the immune system mistakenly attacks healthy cells. These reactions can affect various organs, leading to potentially serious complications [192].

**Table 3 ijms-25-05490-t003:** Traits, comparisons, and synergies of diverse treatments with ICIs.

Treatment	Traits	Comparison with ICIs	Synergies with ICIs	Reference
Surgery	Incision, complications may occur	The excision of malignant neoplasms or tissues does not eradicate residual disseminated cancer cells. While ICIs significantly extend the duration of disease-free survival (DFS) and progression-free survival (PFS) in individuals.	Combination surgery removes the primary tumor and boosts the immune system to enhance ICIs in combating cancer.	[40]
Chemotherapy	Targeted drugs kill fast-growing cancer.	Cancer cells develop resistance to chemotherapy, thus inhibiting patients from achieving a sustained response like ICIs.	Combination therapy is more effective in improving the survival rate and reducing the risk of death. In advanced gastric cancer, CheckMate 649 reduced the risk of death by 20–35% in patients with programmed cell death ligand 1 (PD-L1) CPS ≥ 5.	[8,13]
Radiation therapy	Using high-energy radiation (X-ray) to damage the DNA of the cancer cells directly	RT functions as an inducer by eradicating malignant cells and eliciting a widespread immune reaction, whereas ICIs are monoclonal antibodies designed to block inhibitory molecules expressed on the surface of APCs and CD4+ T cells, thereby modulating the immune system. These two approaches are associated with distinct adverse effects: RT may lead to transient side effects (alopecia, fatigue), while ICIs induce autoimmune and inflammatory responses.	Combination therapy can improve the resistance of tumors to ICIs by increasing cell infiltration of cytokines and simultaneously improving the ME to provide a more favorable environment for ICIs.	[17,18,19,20,25,40]
Targeted therapy	Focuses on targeted proteins that control the growth, division, and spread of individual cancer cells	Targeted therapy is tailored to patients based on the number of specific gene changes found in their cancer cells. However, ICIs do not consider their specific gene changes. For response rates for cancer treatment, targeted therapies cause rapid responses, while ICIs require more time to evaluate. The side effects of both also vary from person to person.	Clinical trials show that targeted therapy combined with ICI can treat various types of cancer, including melanoma, lung cancer, and renal cell carcinoma. It greatly improves anti-tumor effects and enhances survival rates. In addition, this combination therapy has shown great efficacy in improving the prognosis of patients diagnosed with metastatic melanoma.	[14,16,40,193]

## 5. Synergy between Traditional Interventions and ICI Treatment (Table 3)

### 5.1. Surgical Synergy

The combination of ICIs with surgical intervention shows promise for enhancing the effectiveness of cancer treatment. The central objective of this combination treatment is to bolster the immune system’s response to cancer cells [25]. Through excision of the primary tumor and stimulation of the immune system, combination with surgery amplifies the effectiveness of ICIs in combating cancer.

CRS guided by hyperthermic intraperitoneal chemotherapy (HIPEC) is a commonly used surgical treatment for peritoneal metastases, where malignant cells have spread to the lining of the abdominal cavity. The goal of CRS is to remove all visible tumors within the abdominal cavity [194]. Subsequently, heated chemotherapy is administered through HIPEC directly into the affected region to eradicate any residual cancerous cells [195]. Combining surgical and chemical modalities invigorates the immune system and elicits synergism with ICIs, making it a highly effective treatment strategy.

Several empirical studies have shown that combining surgical intervention with immunotherapy through ICIs can benefit patients significantly. For example, a scholarly study published in the Journal of Clinical Oncology in 2020 [196] revealed that combination therapy consisting of surgical intervention and ICIs substantially prolongs the disease progression-free period of patients with metastatic melanoma. Furthermore, a 2019 study published in the Annals of Surgery [197] demonstrated that combining surgical procedures and ICI administration increases complete response rates and prolongs overall survival rates in individuals diagnosed with early-stage lung cancer.

However, combined surgery presents inherent risks and challenges, making it suboptimal for all patients with cancer. Furthermore, the timing and sequencing of combination surgery and ICI administration may impact the effectiveness and safety of this approach. Despite these complex considerations, combined surgery shows immense potential as a groundbreaking treatment option for patients with cancer.

### 5.2. Chemo-Synergy

The combination of chemotherapy and ICIs is a crucial strategy for enhancing the effectiveness of cancer therapy. Chemotherapy can help the immune system identify and destroy cancer cells more effectively and selectively before ICIs are administered [198].

Clinical trials have been conducted to evaluate this concept, and the results have been positive. For example, the KEYNOTE-021 clinical trial examined the use of pembrolizumab in combination with chemotherapy in individuals with NSCLC, and the combined treatment significantly improved survival rates compared to those observed with chemotherapy alone [199]. Additionally, ICIs outperformed the placebo in treating advanced gastric cancer. In the CheckMate 6498 and ORIENT169 trials, ICIs combined with chemotherapy prolonged the overall survival and PFS rates compared to those observed with chemotherapy alone. This combination also reduced the risk of death by 20–35% in patients with a PD-L1 CPS ≥ 5 [200].

In patients with metastatic NSCLC, combining anti-PD-1/PD-L1 mAbs with dual platinum-based chemotherapy has been shown to be more effective than chemotherapy alone. Notably, the study only included patients with totipotent mNSCLC. The findings highlight the potential benefits of combining immune checkpoint inhibitors (ICIs) and chemotherapeutic agents in the treatment of mNSCLC [201].

When considering combination therapy, consulting with a healthcare professional is crucial. It is also important to consider any potential side effects when considering chemotherapy. The proper monitoring and management of these side effects is critical for patient safety. Furthermore, finding the right balance between chemotherapy dosage and ICI administration timing requires further investigation.

### 5.3. Synergy with Targeted Therapy

The integration of targeted therapy has shown promise as a complement to immune checkpoint inhibitors (ICIs) in enhancing the effectiveness of cancer immunotherapy. This combination approach has emerged as a strategy to overcome limitations in the immune system’s ability to recognize and destroy cancer cells. Combining the two approaches can offer a potent strategy to combat cancer.

The amplification of HER2 is a phenomenon that occurs in various tumor types, particularly in mammary glands. Taking the HER2 combination with ICIs as an illustrative case, it has been observed that this phenomenon occurs. HER2 activation leads to the activation of numerous oncogenic signaling pathways, such as the Ras/Raf/ERK pathway, which enhances the survival, proliferation, migration, and resistance of malignant cells to immunotherapy [201]. Therefore, scientific research has recognized it as an emerging objective for breast cancer treatment. It is worth noting that in murine breast cancer cells that express the HER2 receptor, the combination therapy of DS-8201a and an anti-CTLA-4 antibody demonstrated a greater antineoplastic effect compared to monotherapy [202]. In a HER2-expressing localized breast cancer model, co-administration of T-DM1 with anti-CTLA-4/PD-1 ICIs attenuated tumor cell resistance to the ICIs [25].

Several clinical trials have also shown promising results in treating various types of cancer. For example, the combination of the BRAF inhibitor dabrafenib and the MEK inhibitor trametinib has been found to increase the survival rate in patients with BRAF V600 mutation-positive advanced melanoma [203]. Furthermore, this therapy has shown promising results in improving outcomes for patients diagnosed with metastatic melanoma [204]. The combination has demonstrated a significant improvement in progression-free survival and overall survival rates compared to the use of either treatment alone.

### 5.4. Synergy with Radiation Therapy

Radiation therapy has been a long-standing treatment for cancer. It can potentially cause immunogenic cell death, which leads to the release of tumor antigens and proinflammatory signals. This therapy can stimulate the immune system and enhance the effectiveness of ICIs [205]. Additionally, radiation therapy can alter the TME by decreasing immunosuppressive cells and augmenting immunostimulatory cells. Consequently, this phenomenon engenders a more conducive milieu, facilitating ICI efficacy [206].

The combination of radiation therapy and ICIs has also shown promise in other cancers, such as head and neck cancer, NSCLC, and bladder cancer. In a study of head and neck squamous cell carcinoma (HNSCC) in mice, researchers found that malignant tumors were resistant to PD-L1 checkpoint blockade therapy. This resistance was attributed to low levels of PD-L1 expression, inadequate infiltration of tumor-infiltrating lymphocytes, and resistance to radiotherapy. However, the combination of radiotherapy and PD-L1 therapy exhibited a synergistic effect, resulting in the upregulation of PD-L1 expression on tumor cells and increased infiltration of T cells after RT. This effect led to high levels of tumor inflammation, which significantly improved tumor control and ultimately enhanced the survival rate [207]. Furthermore, the PACIFIC trial showed that combining RT with durvalumab can improve the clinical outlook of patients with unresectable stage III NSCLC [208]. In addition, the SWOG/NRG18063 study provided valuable insights into optimal treatment strategies for radiotherapy in muscle-invasive bladder cancer by incorporating atezolizumab [209].

Despite the positive outcomes discussed above, there remain challenges and limitations associated with the combined use of radiation therapy and ICIs. The optimal dose and timing of radiation therapy in relation to ICI administration are still being investigated. Additionally, there may be potential side effects and toxicity associated with this combination therapy, although these have generally been manageable in clinical trials to date.

## 6. In Pursuit of Solutions: Limitations and Challenges in ICI Treatment

### 6.1. Primary and Acquired Resistance to ICI

The concepts of primary and acquired resistance have been proposed to clarify the fundamental mechanisms of resistance to immunotherapy [210]. Acquired resistance arises when the disease escalates following an initial response, whereas primary resistance is generally characterized by cancer cells’ lack of response to immunotherapy. The definition of acquired resistance varies greatly across different demographic groups. Further investigations are required to comprehensively understand acquired and primary resistance, particularly in the context of potential mechanisms and clinical trial design [211].

Numerous factors contribute to primary resistance, including tumor-intrinsic properties. These factors include insufficient tumor neoantigen levels (C17 or f78, HSD17B1, and WNK4) [211], inadequate immune cell infiltration into the TME, and activation of alternative immune checkpoint pathways. Additionally, tumor-extrinsic factors, such as immunosuppressive molecules in the TME, can contribute to primary resistance. Conversely, acquired resistance refers to the development of resistance during or after a successful initial response to ICIs. The mechanisms of acquired resistance include upregulation of the expression of alternative immune checkpoint pathway components, the loss or downregulation of target antigens, mutations in genes involved in antigen processing and presentation, and immune system evasion by tumor cells. These mechanisms can hinder the effectiveness of ICIs, ultimately leading to disease progression [183].

In a randomized, controlled phase III KEYNOTE-006 trial, 834 individuals with advanced melanoma were administered pembrolizumab at a prescribed dose of 10 mg/kg body weight. The drug was administered on either a biweekly or triweekly basis in equal proportions. In that study, the response rate to pembrolizumab dosage every two weeks was 33.7%; that of those treated every three weeks was 32.9%. However, the response rate of patients receiving ipilimumab every three weeks at a dosage of 3 mg/kg was only 11.9%. These observations suggest that primary or acquired resistance may impede the widespread use of immune checkpoint inhibitors in clinical settings [101].

Although phase III clinical trials have shown favorable results, with immune checkpoint inhibitors like ipilimumab increasing the overall survival rate of melanoma patients to approximately 30% for over five years and around 70% for three years, the majority of patients undergoing this treatment still experience a high incidence of melanoma (ranging from 60–70%) and other types of cancer [212].

In conclusion, the effectiveness of immunotherapy in cancer treatment is significantly hindered by primary and acquired resistance. However, current research aims to clarify the complex mechanisms underlying resistance and implement innovative methods to overcome these limitations. The continuous improvement of our understanding of resistance mechanisms can enhance the effectiveness of immunotherapy and maximize its ability to improve patient outcomes in the fight against cancer.

### 6.2. Heterogeneity of Treatment Response

ICIs have emerged as a promising treatment modality for patients with cancer. However, analyzing immune cell subpopulations within tumors is challenging due to the significant heterogeneity of the TME [198]. The TME harbors diverse types of immunosuppressive cells, including myeloid-derived suppressor cells. These cells represent a heterogeneous population of immature cells that are pathologically activated and can efficiently suppress the activity of T cells, contributing to the evasion of immune destruction by malignant tumor cells [213]. Consequently, ICI efficacy is contingent upon the existence of heterogeneity (Figure 3).

Tumor heterogeneity is characterized by variability within a single tumor, diversity within a patient, discrepancies across subtypes of a particular type of cancer, or variations within malignancies originating in disparate tissues. Such heterogeneities make tumors susceptible to an unfavorable response or outright resistance to ICI treatment [214].

Recent research has identified potential biomarkers that may be useful in predicting which patients will respond well to ICI treatment. Mutational allele heterogeneity in tumors serves as an indicator of genetic heterogeneity within tumors and a prognostic biomarker for predicting patient response to medical interventions [215]. Moreover, TMB is a potential biomarker for predicting response rates in certain cancer types [216]. Notably, previous studies related to ICIs have typically relied on the use of isolated single biomarkers to predict the prognosis of patients receiving immunotherapy.

However, the heterogeneity of treatment responses to ICIs presents an obstacle due to the absence of several combined biomarkers for predicting favorable treatment outcomes. Identifying the subset of patients who will benefit from treatment while discerning those who may not poses a challenge for clinicians [217]. Despite the challenges and limitations of ICI treatment, it remains a promising approach for cancer treatment. With further research and development of biomarkers, clinicians may be able to better predict response rates, allowing for more personalized and effective treatments for cancer patients.

### 6.3. Expense Consideration and Access to ICI

Currently, the FDA has approved eight distinct ICIs for treating solid tumors. These include ipilimumab, which functions as an inhibitor of CTLA-4; pembrolizumab, a PD-1 inhibitor; and avelumab, an inhibitor of PD-L1 [218]. It is widely acknowledged that the cost of these medications is considerable. A review of the annual data on healthcare subsidies in the United States between 2011 and 2021 reveals that the total expenditure on ICIs in 2021 was approximately 4.119 billion dollars. This figure represents a remarkable increase of approximately 1470.65 times compared to the expenditure in 2011, which accounted for approximately 2.801 million dollars [219].

A comparative analysis was conducted by researchers to evaluate the therapeutic efficacy of dacarbazine, vemurafenib, or vemurafenib/ipilimumab for the treatment of diseases caused by the BRAF mutation. The study found that both vemurafenib and vemurafenib/ipilimumab were significantly more cost-effective than dacarbazine. However, according to the overall expenses associated with vemurafenib (USD 156,831) and vemurafenib/ipilimumab (USD 254,695), they were significantly higher than those associated with dacarbazine (USD 8391) [220]. In 2017, McCrea et al. [221] administered nivolumab in conjunction with ipilimumab to address recurrent RCC as an example of the management of genitourinary cancers (GUCs) using ICIs. The investigators discovered that the cost of nivolumab was higher than that of ipilimumab, amounting to USD 151,676 per quality-adjusted life year (QALY) [222]. These high costs and long-term investments do not provide cost-effectiveness.

Although ICIs have achieved significant success in treating certain diseases, including tumors and cancers, either alone or in combination therapies, their accessibility remains limited, and they face several challenges. The cost of treating various diseases with ICIs can be exorbitant, making it unfeasible for some individuals to access this treatment option. This financial burden disproportionately affects low-income families, limiting their access to ICIs. In addition, inadequate medical technology in underdeveloped regions creates numerous obstacles to obtaining ICIs.

### 6.4. Patient Selection and Identification of an Optimal Combination

The identification of optimal combination treatments and patient selection are crucial to the success of ICIs. Due to the complexity of the immune system and the nature of each cancer, it is essential to accurately determine the ideal combination of ICIs. Patient selection also involves identifying individuals who are more likely to experience irAEs. The main agents used for tumor immunotherapy are activated T lymphocytes. Although ICIs can potentially enhance T-cell-mediated tumor immunity, their use also carries the risk of damaging healthy tissue cells, resulting in irAEs [223]. Clinicians must closely monitor and manage potential complications to identify patients at greater risk of such events, ensuring patient safety.

To illustrate, the immuno-oncology combination protocol aims to evaluate the effectiveness of anti-PD-1/anti-PD-L1 drugs in a preliminary trial of immunotherapy combinations for hepatocellular carcinoma (HCC). In 2018, Pishvaian et al. [224] administered atezolizumab (1200 mg) and bevacizumab (15 mg/kg) once every three weeks to patients with HCC. The results revealed an overall response rate (ORR) of 32% (1 complete response, 22 partial responses) and a median PFS duration of 14.9 months. In 2019, Xu et al. [225] reported that treating patients with SHR-1210 (200 mg) every two weeks and apatinib daily resulted in an ORR of 50% (8 partial responses) and a median PFS duration of 5.8 months [226]. These findings highlight the importance of continuously investigating drug dosages in combination therapy and selecting drug combinations to optimize patient benefits.

However, although these efforts have somewhat alleviated the challenges surrounding patient selection and the identification of optimal combination treatments for ICIs, much work is still needed. An improved understanding of the immune system, cancer biology, and mechanisms of ICIs will play a key role in the continued development of this field.

The application of ICIs in cancer immunotherapy shows promise; however, patient selection and the identification of effective combination treatments remain challenging. Further research and development efforts are necessary to overcome these constraints and ensure continued progress.

## 7. Conclusions

Unprecedented progress has been achieved recently in the development of ICIs for the treatment of various cancers. ICIs regulate appropriate immune responses and promote tumor cell evasion of immune clearance within the tumor microenvironment. The PD-1 pair, CTLA-4, and CISH proteins represent three key ICI approaches discussed in this review. Additionally, recent research has highlighted other promising immune checkpoints, such as LAG-3, TIM-3, and TIGIT, which exhibit distinct regulatory functions and may serve as potential targets for future therapeutic intervention. ICIs are increasingly being used to treat melanoma, lung cancer, renal cell carcinoma, bladder cancer, and lymphoma. Considering the specific characteristics of different tumors, ICIs and special combination strategies have been selected and continuously tested in clinical trials. Several of these strategies have demonstrated broader applicability and more durable responses in clinical treatment, indicating a promising future for such approaches. Nevertheless, it is crucial to acknowledge that while ICIs have demonstrated remarkable efficacy in specific tumor types, such as melanoma and NSCLC, their efficacy in other cancers remains limited. This finding underscores the necessity for further research to identify predictive biomarkers and optimize treatment regimens for these indications. Various combinations of ICIs with surgery, chemotherapy, or targeted therapies have demonstrated promising effects in clinical trials. These combination approaches overcome the disadvantages of surgery, which cannot eliminate tumor cells that migrate to other sites, and the indifferent killing of normal cells by chemotherapy. Targeted therapy, a subset or application of precision medicine, has emerged as a prominent topic in the current era. Targeted therapy is based on specific gene alterations, whereas ICI therapy is focused on enhancing intrinsic immune responses to eliminate tumor cells. Recent advances in targeted therapy have led to the development of novel small-molecule inhibitors and antibody–drug conjugates targeting oncogenic drivers such as EGFR, ALK, and BRAF. These agents have demonstrated synergistic effects when combined with ICIs in preclinical and clinical studies. Integrating these innovative approaches holds great promise for improving the efficacy of immunotherapy against cancer. It is also important to acknowledge the limitations of ICIs, which include the development of resistance, significant side effects, heterogeneity of use, and high costs. Although a significant number of patients treated with ICIs have demonstrated increased overall survival rates, a high recurrence rate of eliminated or other tumors has been reported in these patients. Moreover, the long-term safety and efficacy of ICIs remain a concern, particularly regarding irAEs, such as autoimmune toxicities, which can lead to treatment discontinuation and compromise patient outcomes. IrAEs, resulting from nonspecific activation of the immune system, represent a notable consequence of ICI therapy. These events are manifested by the blockade of CTLA-4, PD-1, or PD-L1, which results in uncontrolled immune activation and subsequent tissue damage. The severity of irAEs is highly variable, ranging from relatively mild dermatitis to life-threatening colitis or pneumonitis. While irAEs present significant challenges in ICI therapy, their incidence demonstrates the potent immunomodulatory effects of these agents. Further research is necessary to elucidate the underlying mechanisms of irAEs and develop strategies for their prevention and management without compromising the antitumor efficacy of ICIs. In conclusion, while ICIs offer significant promise in cancer treatment, ongoing research is essential to address their limitations and optimize their use. To maximize the potential of ICIs for improving cancer outcomes, it is essential to balance the benefits and disadvantages of ICIs and employ personalized approaches according to patient characteristics.

## Figures and Tables

**Figure 1 ijms-25-05490-f001:**
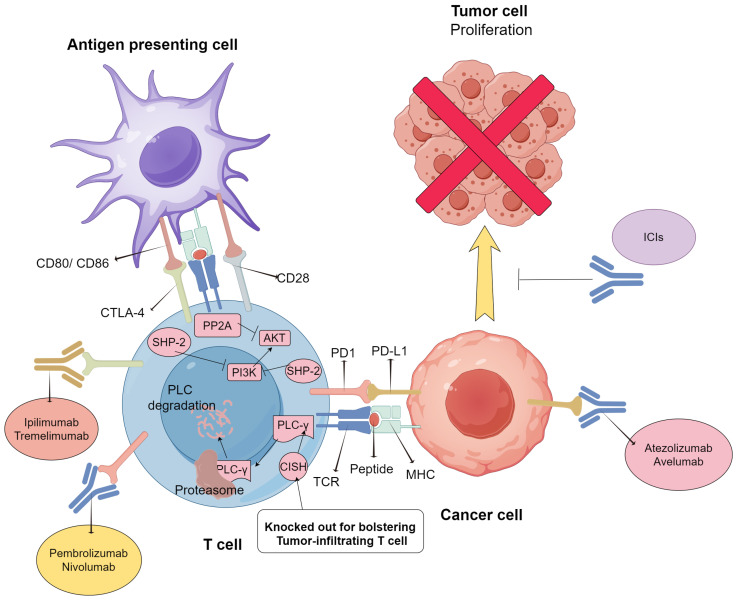
Mechanisms of action of PD-1/PD-L1, CTLA-4, and CD80/CD86, and the regulating role of CISH in T cells’ activities. CD80/86 expressed on APCs or tumor cells interacts with CTLA-4 and its homologous CD28, or PD-L1 interacts with its receptor PD1 on the surface of the activated T cells, respectively. Such interactions result in the activation of SHP2 and PP2A and the downregulation of the PI3K/AKT axis. CISH could regulate T cell activities via the lysis of PLC-γ by the proteasome, but not via PI3K/AKT axis. CISH knockout could enhance tumor-infiltrating T cell activity, especially when combined with PD1 inhibitors. ICIs targeting different molecules are also listed. Abbreviations: TCR: T-cell receptor; PD1: programmed cell death protein-1; PD-L1: programmed cell death protein ligand 1; CTLA-4: cytotoxic T-lymphocyte-associated protein 4; MHC: major histocompatibility complex; SHP-2: Src homology region 2-containing protein tyrosine phosphatase 2: AKT: protein kinase B; PI3K: phosphatidylinositol 3-phosphokinase; PP2A: protein phosphatase 2A; PLC-γ: phospholipase C; CISH: cytokine-inducible SH2-containing protein.

**Figure 2 ijms-25-05490-f002:**
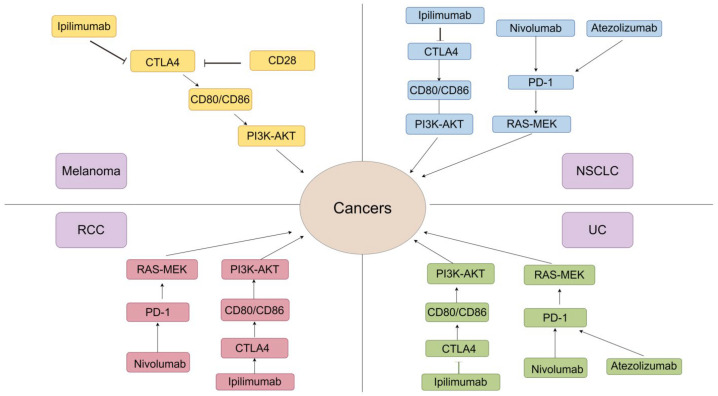
Molecular mechanisms of ICIs in various cancers. The application of ICIs in these five cancers has reached a very advanced level. In terms of molecular mechanisms, ICIs are mainly mediated by PD-1 and CTLA-4, but the effectiveness of the same drug varies in different cancers. Abbreviations: NSCLC: non-small cell lung cancer; UC: urothelial carcinoma; RCC: renal cell carcinoma.

**Figure 3 ijms-25-05490-f003:**
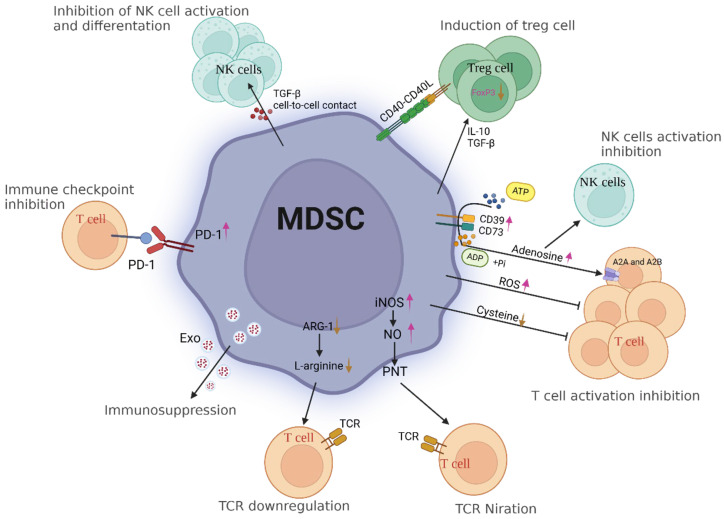
Immunosuppressive dynamics in the tumor microenvironment (TME): MDSC-mediated modulation of immune responses. Myeloid-derived suppressor cells (MDSCs) are a heterogeneous population of immune cells that play a significant role in the tumor microenvironment. In cancer, MDSCs are induced and accumulate in response to factors released by tumor cells. Excessive production of NO by iNOS in MDSCs can suppress T-cell function and promote immune evasion. NO inhibits T-cell receptor signaling by nitrosylating critical signaling proteins, reducing IL-2 production, and impairing cytotoxic activity. This dampens T cell effector functions and promotes an immunosuppressive environment. Furthermore, PNT, which is formed by the reaction between NO and superoxide radicals, has been implicated in inducing apoptosis or dysfunction of activated T cells. MDSCs secrete immunosuppressive cytokines such as IL-10 and TGF-ß. These cytokines can modulate cysteine metabolism within the tumor microenvironment, promoting an immunosuppressive state that inhibits T-cell responses. CD39 is an ectonucleotidase that hydrolyzes ATP into AMP and subsequently into ADP; CD73 converts AMP into adenosine. This further promotes the accumulation of adenosine in the tumor microenvironment or sites of inflammation. Adenosine binds to specific G-protein-coupled receptors on the surface of T cells, particularly the AZA and A2B adenosine receptors. Activation of these receptors inhibits T-cell receptor signaling pathways and downstream activation events, thereby suppressing T-cell activation, proliferation, and cytokine production. ARG-1 competes with T cells for the essential amino acid L-arginine, leading to its depletion in the tumor microenvironment or inflammatory sites. L-arginine is necessary for T-cell activation, proliferation, and effector functions. Therefore, decreased availability of L-arginine can impair T-cell responses. PD-1 is an immune checkpoint receptor expressed on the surface of activated T cells. When PD-1 binds to its ligands, such as PD-L1 or PD-L2, which are expressed by MDSCs and other immune cells, it delivers inhibitory signals that attenuate T-cell activation and effector functions. TGF-ß signaling can directly inhibit the activation and cytotoxicity of NK cells. It dampens NK cell receptor-mediated signaling pathways, such as those initiated by activating receptors like NKG2D or DNAM-1, which are essential for NK cell recognition and the killing of target cells. Both IL-10 and TGF-ß can promote the differentiation and expansion of regulatory T cells. They contribute to the generation of a population of immunosuppressive Tregs that can suppress excessive immune responses. Abbreviations: MDSCs: myeloid-derived suppressor cells; NO: nitric oxide; iNOS: inducible nitric oxide synthase; PNT: peroxynitrite.

**Table 1 ijms-25-05490-t001:** Information about ICIs in different clinical trials.

Cancer	Agents	Study/Refs.	Dose	ORR
Melanoma	Pembrolizumab (PD-1 inhibitor)	NCT02821000 (phase 1 study, N = 103)	2 mg/kg/Q3W	16.7%
	Ipilimumab (CTLA-4 inhibitor)	NCT01515189 (phase 3 study, N = 727)	10 mg/kg/Q3W	15%
			3 mg/kg/Q3W	12%
	Nivolumab (PD-1 inhibitor)	NCT00730639 (phase 1 study, N = 395)	0.1 mg/kg/Q2W	35.3%
			0.3 mg/kg/Q2W	27.8%
			1.0 mg/kg/Q2W	31.4%
NSCLC	Pembrolizumab (PD-1 inhibitor)	NCT03515837 (phase 3 study, N = 492)	Pembrolizumab 200 mg/Q1W + Pemetrexed + Chemotherapy	5.6%
			Placebo 200 mg/Q1W + Pemetrexed + Chemo	5.5%
	Nivolumab + ipilimumab (PD-1 inhibitor + CTLA-4 inhibitor)	NCT02864251 (phase 3 study, N = 367)	Nivolumab 360 mg/Q3W + Chemotherapy	19.35%
			Nivolumab 3 mg/kg/Q2W + Ipilimumab 1 mg/kg/Q6W	17.12%
			Platinum Doublet Chemotherapy	15.90%
RCC	Nivolumab (PD-1inhibitor)	NCT01354431 (phase 2 study, N = 168)	0.3 mg/kg IV Q3W	18.45%
			2 mg/kg IV Q3W	25.46%
			10 mg/kg IV Q3W	24.82%
UC	Pembrolizuma (PD-L1 inhibitor)	NCT03361865 (phase3 study, N = 93)	Pembrolizumab 200 mg/Q3W + Epacadostat 100 mg BID	31.8%
			Pembrolizumab 200 mg/Q3W + Placebo BID	24.5%
HNSCC	Pembrolizumab (PD-1 inhibitor)	NCT01848834 (phase 1b study, N = 192)	10 mg/kg/Q2W	18%
			200 mg/Q3W	

**Table 2 ijms-25-05490-t002:** ICI types, their targets, and their major applications in tumors.

Category	Target Molecule	FDA-Approved Drugs	Applications	Reference
PD1 + PD-L1 inhibitor	PD1	Nivolumab, Pembrolizumab	NSCLC Renal cell carcinoma Hodgkin’s lymphoma	[13,25,26]
	PD-L1	Avelumab, Atezolizumab, Durvalumab	Merkel cell carcinoma Urothelial cancer NSCLC	[27,28,29]
CTLA-4 inhibitor	CTLA-4	Ipilimumab Tremelimumab	Melanomas Renal cell carcinoma Hepatocellular carcinoma	[30,31,32]
CISH	PLC-γ1	CISH-Knocked tumor-infiltrating T cell	Melanomas Gastrointestinal carcinoma	[33]

## Data Availability

Not applicable.

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
