# Peer review of "Immunomodulatory Precision: A Narrative Review Exploring the Critical Role of Immune Checkpoint Inhibitors in Cancer Treatment"

_ijms, 2024, doi:10.3390/ijms25105490_

Round 1

Reviewer 1 Report

Comments and Suggestions for Authors

Article No ijms-2955522 “Immunomodulatory Precision: Exploring the Critical Role of Immune Checkpoint Inhibitors in Cancer Treatment” for International Journal of Molecular Sciences

Comments:

The main question addressed in the paper is the usage of immune checkpoint inhibitors (ICIs) in fight against cancers of different origin. To present its advantages but also limitations. It is quite interesting approach, worth presenting and exploring.

Generally it is not new data, but Authors tried to present the issue in an alternative manner. The strong point of the paper is to show both good and limited side of the ICIs usage.

The paper is written well and the language is clear enough to easily embrace the material. The conclusion, in my opinion, answer the question posed at the initial stage of the paper. However, it would be advisable to present ICIs’ effectiveness in primary and metastatic tumors.

1.     In Paragraph 3, the authors focused mainly on selected solid tumors. Please also include a subsection on the role of Immune Checkpoint Inhibitors in Acute Myeloid Leukemia or Lymphomas.

2.     Please change the name of Paragraph 7 from "Discussion" to "Conclusions". The second name more closely reflects the summary meaning of this chapter.

Author Response

Response to Reviewers’ Comments

We thank the reviewers for the helpful comments and suggestions. The comments and suggestions are all valuable for revising and improving our review. We have studied the comments carefully and have added some information to the revised manuscript.

Reviewer #1: 

  1. In Paragraph 3, the authors focused mainly on selected solid tumors. Please also include a subsection on the role of Immune Checkpoint Inhibitors in Acute Myeloid Leukemia or Lymphomas.

RE: We apologize that we only discussed the selected solid tumors in the manuscript. After considering the reviewer’s comments, we added a paragraph of the role of ICIs in lymphomas to fill the gap in this review. We have also made changes to the corresponding figures on this section.

  1. Please change the name of Paragraph 7 from "Discussion" to "Conclusions". The second name more closely reflects the summary meaning of this chapter.

RE: In the previous "Discussion" part, we mainly summarized the whole manuscript, but there is less critical thinking and analysis of our findings in this review. At the very beginning, the limitations and challenges of ICIs have been concluded in the discussion part. Given that, the limitations and challenges of ICIs are well worth discussing, however, and these cannot be explained in a few words. As a result, we decided to make them a separate paragraph. After combining with the reviewers' recommendations, we changed the name of Paragraph 7 from "Discussion" to "Conclusions".

Reviewer 2 Report

Comments and Suggestions for Authors

Immune checkpoint inhibitors have significantly changed the treatment of cancer patients in recent years. In 'Immunomodulatory Precision: Exploring the Critical Role of Immune Checkpoint Inhibitors in Cancer Treatment', the authors provide a comprehensive overview of the classification, mechanism of action, use, and combination strategies of ICIs in various cancers and discuss their current limitations. The manuscript is well-written and presented in a well-structured manner. The cited references are mostly recent publications and relevant. The topic taken up by the authors is important and topical. However, in my opinion, the content of the article is too broad. I also suggest that the authors formulate the research question, extract keywords, and inclusion and exclusion criteria, and provide information on the databases used and search strategies. It would also be beneficial to separate the conclusion section.

More specific comments:

1. Authors should include a research question in the manuscript.

2. In recent years, inhibition of the activity of negative immune checkpoints has become a very promising strategy for cancer treatment. The most commonly used are CTLA-4 receptor blockade and blockade of binding between PD-1 and PD-L1 molecules. The efficacy of monoclonal antibodies directed against immune checkpoints inhibiting the immune response has been confirmed in the treatment of cancers such as malignant melanoma, non-small cell lung cancer, renal cell carcinoma, colorectal cancer and bladder cancer, neuroendocrine skin cancer or Hodgkin's lymphoma. The best results were obtained in the treatment of melanoma. Unfortunately, the use of anti-CTLA-4 and anti-PD-1 monoclonal antibodies in combination therapy, although relatively effective, is also associated with significant toxicity. In addition, the problem of the insensitivity of a significant proportion of patients to this therapy remains unresolved. To overcome these barriers, treatments combining checkpoint inhibitors with other anti-cancer therapies are being used. Recently, clinical trials have been conducted that test the use of checkpoint inhibitors alongside traditional therapies. This article reviews the classification, mechanism of action, use and combination strategies of ICIs in various cancers while discussing their current limitations. Presenting the topic in this way provides a comprehensive overview of the research area under discussion.

3. This review discusses the mechanisms of immune checkpoint inhibitors (ICIs) in tumor suppression and their use in treating various cancers. Additionally, this review considers the existing constraints of ICIs and presents novel insights into their developmental prospects. However, as I have highlighted, the article is too broad and therefore difficult to read. The authors should emphasize the current progress of knowledge in the research area under consideration.

4. As I pointed out in the review, authors should, in addition to the aforementioned formulation of the research question, extract keywords, inclusion and exclusion criteria and provide information on the databases and search strategies used.
As this is a review article, it is difficult to suggest that the authors supplement the manuscript with specific control methods. They could, however, discuss directions for further research in greater detail.

5.  There are no conclusions in the manuscript, so I suggested to the authors to separate the conclusions section.

6. The references are appropriate.

7. The tables and figures are of good quality and prepared in accordance with the “Instructions for Authors”.

Author Response

Reviewer #2: 
1. Authors should include a research question in the manuscript.

RE: We apologize for the lack of a research question. Considering the experts' suggestions, we added a research question to the introduction to emphasize the central point of our review. We also made minor revisions to the introduction and reiterated points to make our points more understandable to readers and reviewers.

Reviewer 3 Report

Comments and Suggestions for Authors

Thank you for giving me this opportunity to review the review paper entitled, Immunomodulatory precision: exploring the critical role of immune checkpoint inhibitors in cancer treatment. I here carefully reviewed the submitted set of manuscript and found it merits for publication following some revisions as described below.

1. This is a narrative review, should add a description to the title.

2. The irAE, adverse events on such immune checkpoint inhibitors in cancer treatment should also described in each inhibitor and the Discussion section.

Comments on the Quality of English Language

English grammatical checkings should be recommended.

Author Response

Response to Reviewers’ Comments

We thank the reviewers for the helpful comments and suggestions. The comments and suggestions are all valuable for revising and improving our review. We have studied the comments carefully and have added some information to the revised manuscript.

Reviewer 3#:

3: This review discusses the mechanisms of immune checkpoint inhibitors (ICIs) in tumor suppression and their use in treating various cancers. Additionally, this review considers the existing constraints of ICIs and presents novel insights into their developmental prospects. However, as I have highlighted, the article is too broad and therefore difficult to read. The authors should emphasize the current progress of knowledge in the research area under consideration.

4: As I pointed out in the review, authors should, in addition to the aforementioned formulation of the research question, extract keywords, inclusion and exclusion criteria and provide information on the databases and search strategies used. As this is a review article, it is difficult to suggest that the authors supplement the manuscript with specific control methods. They could, however, discuss directions for further research in greater detail.

RE: We apologize that our review is too broad and incomprehensible mainly because we have read and thought a lot about the relevant literature in the field of ICIs. As a result, we finally wrote this review and covered a wide range of topics. By combining the expert’s suggestions, we do the following revision:

We deleted some parts of repetitive, unnecessary, and outdated information and reorganized some paragraphs to help readers and experts understand the manuscript and enable further access to cutting-edge knowledge of this field.

On lines 193-198, we deleted “The expression of PD-L1 on the tumor surface is strongly correlated with poor T cell immune clearance. Therefore, the detection of PD-L1 in TME can predict the effectiveness of anti-PD treatment. Among the various tumors expressing PD-L1, lymphoma is the most responsive to anti-PD therapies. Patients with relapsed or refractory Hodgkin's lymphoma treated with nivolumab showed an overall response rate of more than 80%” in original manuscript.

On lines 160-162, we deleted “Antibodies designed to inhibit the PD-1/PD-L1 axis have shown efficacy in treating various solid tumors and hematologic malignancies, including melanomas, Hodgkin’s lymphomas, and NSCLC, as demonstrated in over 1000 clinical trials” in original manuscript.

On lines 179-182, we deleted “The initial indication that PD-1 hinders the immune surveillance of tumor cells emerged from the observation that the overexpression of PD-L1 on P815 tumor cells significantly suppressed the cytolytic activity of CD8+ T cells” in original manuscript.

On lines 183-186, we deleted “Tumor cells that express PD-L1 communicate with PD-1-positive T cells, resulting in T cell exhaustion or apoptosis. This process enables the evasion of cytotoxic T lymphocyte (CTL) lysis” in original manuscript.

On lines 202-203, we deleted “The primary function of ICIs is to reinvigorate exhausted T cells and induce normal immune responses” in original manuscript.

On lines 212-215, we deleted “Two examples that illustrate the “high burden for better predictions” are lung cancer and melanoma, both cancers have a high number of mutations caused by cigarette smoking and UV exposure, respectively, and respond well to anti-PD therapies” in original manuscript.

On lines 240-242, we deleted “Although the mechanisms of CTLA-4 and PD-1 involved ICI therapies are different, they both are co-inhibitory molecules, thus equally important for cancer treatment” in original manuscript.

On lines 261-262, we deleted “Furthermore, CTLA-4 on Treg cells stabilizes the interaction between regulatory and conventional T cells to suppress the latter” in original manuscript.

On lines 268-270, we deleted “Therefore, by blocking CTLA-4 in tumor cells, the immune system is no longer suppressed, and the mechanisms against tumor cells are reactivated” in original manuscript.

In our original manuscript, lines 215 was re-established from the beginning of “the success of combinating……” to the end of this paragraph.

On lines 152-153, we deleted “Our recent focus has been on identifying more sensitive biomarkers, raising response rates, and eliminating negative reactions.” in original manuscript.

On lines 155-159, we deleted “The TME is where T cell-tumor interactions occur, and immune checkpoints generate inhibitory signals to repress excessive immune reactions and autoimmunity. However, tumor cells can exploit this process to suppress the immune response, resulting in significant proliferation. ICIs leverage the body's immune system to eliminate neoplastic lesions by modulating the immune response.” in original manuscript.

On lines 166-168, we deleted “The role of PD-1 is to inhibit T cell proliferation and activation, alter T cell metabolism and cytokine secretion, and ultimately induce apoptosis of activated T cells” in original manuscript.

On lines 174-179, we deleted “PD-1 ligands, mainly PD-L1 and to a lesser extent PD-L2, are expressed not only by normal APCs but also by tumor cells and stromal cells. The expression of these ligands can be regulated by primary mechanisms (non-immune-driven) or secondary mechanisms (immune-driven). The former includes genetic-driven, microRNA-based control and oncogene expressions, while the latter is regulated by signal molecules produced by immune cells in TME” in original manuscript.

On lines 149-152, we added “A research team has confirmed that the PD-L1-related miRNA profile has the potential to forecast the response of Lung Squamous Cell Carcinoma (LUSC) patients to PD-L1/PD-1 inhibitors, aiding in the identification of the optimal treatment cohort.” in the current manuscript.

On lines 146-166, we added “Additionally, research articles have reported that tumor-associated macrophages (TAMs), as major components of the TME, significantly impact the therapeutic efficacy of PD-1/PD-L1 inhibitors.” in the current manuscript.

On lines 188-201, we added “PD-L1 expression and tumor mutational burden are common predictors of response to anti-PD-1/PD-L1 therapy in lung cancer, but other factors like tumor-specific genes, dMMR/MSI, and the gut microbiome also show promise as predictive biomarkers. Non-invasive peripheral blood biomarkers, including DNA-related biomarkers and hematological cell-related biomarkers, are being utilized to predict immunotherapy response. Research has shown that the gut microbiota plays a significant role in the efficacy of anti-PD-1/PD-L1 antibodies in colorectal cancer (CRC)patients, with evidence suggesting that altering the gut microbiota composition can enhance the anti-bodies' effectiveness. Meanwhile, research shows that the strength of the bond be-tween chimeric antigen receptor (CAR) and its target antigen impacts how responsive CAR-T cells are to inhibition by PD-1/PD-L1. The PD-1/PD-L1 axis inhibits T-cells in CAR-T cell therapy for solid tumors, but disrupting this pathway is complex. The study highlights that the affinity between CAR and its antigen is crucial in determining how susceptible T-cells are to PD-1/PD-L1 inhibition, particularly in diseases like lung cancer” in the current manuscript.

On lines 206-209, we added “Neoadjuvant PD-(L)1 inhibitors are safe and effective for treating muscle invasive bladder cancer. Combining them with other immune checkpoint inhibitors and chemotherapy can lead to higher response rates, but also more severe side effects.” in the current manuscript.

On lines 252-256, we added “The activation of Treg cells with CTLA-4-independent immunosuppression can im-pair anti-tumor immunity mediated by CTLA-4 blockade. The depletion of Treg cells in the TME using anti-CTLA-4 mAbs with antibody-dependent cell-mediated cytotoxicity (ADCC) activity is considered a key mechanism to achieve tumor regression.” in the current manuscript.

On lines 270-277, we added “In clinical studies, CTLA4 deficiency rescued functional T cells in leukemia patients who had failed previous CAR T-cell therapy. Thus, selective CTLA4 deficiency may re-vitalize functionally impaired T cells in chronic lymphocytic leukemia (CLL) patients, providing a strategy to enhance patient responsiveness to CAR T-cell therapy. Combination of anti-PD-1 and anti-CTLA-4 checkpoint blockades: treatment of melanoma. However, combination checkpoint inhibition also poses significant clinical challenges and is associated with an increased incidence of immune-related adverse events. ” in the current manuscript.

On lines 291-295, we added “CISH expression is also associated with longer metastasis-free interval (MFI) in triple-negative breast cancer (TNBC) and can refine the prognostic value of PD-L1 expression. This finding may underscore the clinical relevance of combining CISH inhibition with anti-PD-1/PD-L1 in the current manuscript.

On lines 322-330, we added “Additionally, researchers demonstrate a critical role for CIS in suppressing natural killer (NK) cell–mediated control of tumor initiation and metastasis. Mice lacking CISH are highly resistant to methylcholanthrene-induced sarcoma and prevent lung metastasis of B16F10 melanoma and RM-1 prostate cancer cells. Combining CISH knockout with targeted therapies such as BRAF and MEK inhibitors, immune checkpoint blockade antibodies, IL-2, and type I interferons reveals further control of metastasis. These data suggest that targeting CIS can enhance the anti-tumor function of NK cells, and CIS holds great promise as a new target for NK cell immunotherapy.” in the current manuscript.

  1. There are no conclusions in the manuscript, so I suggested to the authors to separate the conclusions section.

RE: As the discussion part is more likely to be the conclusion of our manuscript and incorporating the comments of the first reviewer, we finally decided to change the section “discussion” to “conclusions”.

Reviewer #3: 

  1. This is a narrative review, should add a description to the title.

RE: We added the description “narrative review” to the title and changed it to “Immunomodulatory Precision: A Narrative Review Exploring the Critical Role of Immune Checkpoint Inhibitors in Cancer Treatment”

  1. The irAE, adverse events on such immune checkpoint inhibitors in cancer treatment should also described in each inhibitor and the Discussion section.

RE: We added “PD-1 and PD-L1 related irAE” at the end of section 2.2.1 and “CLTA-4 related irAE” at the end of section 2.2.2. Also, irAE has been described in the “Discussion” section.

Round 2

Reviewer 2 Report

Comments and Suggestions for Authors

The authors have mostly revised their manuscript. Although they have not included information on the databases used and search strategies in the text I recommend the article for further proceedings in its current form.